# Diffusiophoresis of a Soft Particle as a Model for Biological Cells

Hiroyuki Ohshima 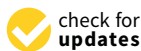

Faculty of Pharmaceutical Sciences, Tokyo University of Science, 2641 Yamazaki, Noda 278-8510, Japan; ohshima@rs.noda.tus.ac.jp

**Abstract:** We derive the general expression for the diffusiophoretic mobility of a soft particle (i.e., polyelectrolyte-coated hard particle) in a concentration gradient of electrolytes for the case in which the particle's core size is large enough compared with the Debye length. Therefore, the particle surface can be regarded as planar, and the electrolyte concentration gradient is parallel to the core surface. The obtained expression can be applied for arbitrary values of the fixed charge density of the polyelectrolyte layer and the surface charge density of the particle core. We derive approximate analytic mobility expressions for soft particles of three types, i.e., (i) weakly charged soft particles, (ii) soft particles with a thick polyelectrolyte layer, in which the equilibrium electric potential deep inside the polyelectrolyte layer is equal to the Donnan potential, and (iii) soft particles with an uncharged polymer layer of finite thickness.

**Keywords:** diffusiophoretic mobility; diffusiophoresis; soft particle



## 1. Introduction

Diffusiophoresis is the migration of colloidal particles in a concentration gradient of electrolytes. A great number of theoretical studies have been reported on the diffusiophoresis of rigid particles [1–16], liquid drops [17–20], and soft particles [21–26]. In particular, soft particles (i.e., polyelectrolyte-coated particles) serve as a model for biocolloids, including biological cells [27–31]. There are several experimental studies on the diffusiophoresis of biological cells [32,33]. In addition, as Shin [34] has pointed out, there is a growing interest in microfluidic colloid separation enabled by diffusiophoresis for various biological and biomedical applications.

In a previous paper [26], we treated a weakly charged soft particle. In the present paper, we derive the general expression and approximate expressions for the diffusiophoretic mobility of a soft particle applicable for arbitrary values of the fixed charge density of the polyelectrolyte layer and the surface charge density of the particle core. We consider soft particles for the case in which the particle's size is large enough compared with the Debye length, $1/\kappa$. Therefore, the particle surface can be regarded as planar. This condition holds for biological systems, since the size of biological cells is of the order of 1~10 μm and $1/\kappa \approx 1$ nm under physiological conditions. We also confine ourselves to the case in which the polyelectrolyte layer is much thinner than the particle core size; therefore, the electrolyte concentration gradient is parallel to the core surface. We derive approximate mobility formulas, which cover the following three important cases: (i) weakly charged soft particles, (ii) soft particles with a thicker polyelectrolyte layer than the Debye length and the Brinkman length, in which the equilibrium electric potential far inside the polyelectrolyte layer is given by the Donnan potential, and (iii) soft particles with an uncharged polymer layer of finite thickness.

## 2. Theory

Let us consider a soft particle, i.e., a hard particle covered with an ion-penetrable surface layer of polyelectrolytes with a thickness $d$ and with a diffusiophoretic velocity $U$

in an aqueous liquid solution of viscosity $\eta$ and of relative permittivity $\varepsilon_r$ containing an electrolyte under a constant applied gradient of electrolyte concentration (Figure 1). The electrolyte is of the Z:Z symmetrical type with valence $Z$ but may have different ionic drag coefficients $\lambda_+$ and $\lambda_-$ for cations and anions, respectively. We assume that the polyelectrolyte layer, in which charges are distributed with a uniform density $\rho_{fix}$ (when $\rho_{fix} = 0$, the surface layer consists of uncharged polymers). We adopt the Brinkman–Debye–Bueche model [35,36]. In this model we regard polymer segments in the polyelectrolyte layer as resistance centers, exerting frictional forces on the liquid flowing in the polyelectrolyte layer. The $x$-axis is considered to be normal to the particle core with its origin $x = 0$ at the front edge of the polyelectrolyte layer, and the $z$-axis is considered to be parallel to the particle core, as shown in Figure 1.

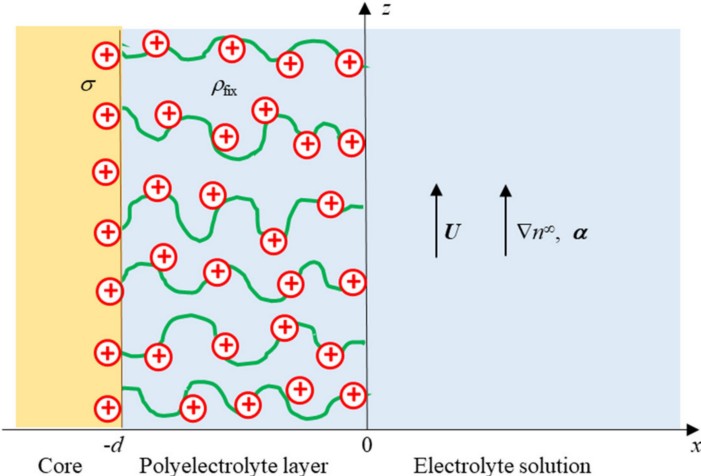

**Figure 1.** Soft particle consisting of the core covered with an ion-penetrable surface layer of polyelectrolytes of thickness $d$. The particle moves with diffusiophoretic velocity $U$ in an electrolyte concentration gradient $\nabla n^\infty$ or with the corresponding vector $a$. $U$ is parallel to $\nabla n^\infty$ and $\alpha$. $\rho_{fix}$ is the fixed charge density in the polyelectrolyte layer, and $s$ is the surface charge density of the particle core.

We consider first the case in which there is no electrolyte concentration gradient. With the functions $n_+(x)$ and $n_-(x)$, which are functions of only $x$, we can denote the concentrations (number densities) of electrolyte cations and anions, respectively, and with $n^\infty$, we can denote their concentration beyond the electrical double layer around the particle, where $n_+(\infty) = n_-(\infty) = n^\infty$. We assume that in the absence of the electrolyte concentration gradient the equilibrium concentrations $n_+(x)$ and $n_-(x)$ of cations and anions, respectively, which depend only on $x$, obey the Boltzmann distribution, viz.,

$$n_\pm(x) = n^\infty e^{\mp y} \tag{1}$$

and the electric potential $\psi(x)$, which also depends only on $x$, satisfies the Poisson–Boltzmann equations:

$$\frac{d^2 y}{dx^2} = \kappa^2 \sinh y \quad \text{for } x > 0 \tag{2}$$

$$\frac{d^2 y}{dx^2} = \kappa^2 (\sinh y - \sinh y_{DON}) \quad \text{for } -d < x < 0 \tag{3}$$

with

$$\kappa = \sqrt{\frac{2Z^2 e^2 n^\infty}{\varepsilon_r \varepsilon_o kT}} \tag{4}$$

$$y(x) = \frac{Ze\psi(x)}{kT} \tag{5}$$

$$y_{\text{DON}} = \frac{Ze\psi_{\text{DON}}}{kT} \tag{6}$$

$$\psi_{\text{DON}} = \frac{kT}{Ze}\text{arcsinh}\left(\frac{\rho_{\text{fix}}}{2Zen^\infty}\right) = \frac{kT}{Ze}\ln\left[\frac{\rho_{\text{fix}}}{2Zen^\infty} + \sqrt{\left(\frac{\rho_{\text{fix}}}{2Zen^\infty}\right)^2 + 1}\right] \tag{7}$$

where $\varepsilon_{\text{o}}$ is the permittivity of a vacuum, $\kappa$ is the Debye–Hückel parameter, $1/\kappa$ is the Debye length, $y(x)$ is the scaled electric potential, $\psi_{\text{DON}}$ is the Donnan potential in the polyelectrolyte layer, and $y_{\text{DON}}$ is the scaled Donnan potential. The boundary conditions for $y(x)$ are

$$\psi(x) \to 0 \ \text{ as } x \to \infty \tag{8}$$

$$\psi(0^-) = \psi(0^+) \tag{9}$$

$$\left.\frac{d\psi}{dx}\right|_{x=0^-} = \left.\frac{d\psi}{dx}\right|_{x=0^+} \tag{10}$$

$$\left.\frac{d\psi}{dx}\right|_{x=d^+} = -\frac{\sigma}{\varepsilon_{\text{r}}\varepsilon_{\text{o}}} \tag{11}$$

where $\sigma$ is the surface charge density of the particle core.

　　Now, we consider the case in which the electrolyte concentration gradient $\nabla n^\infty$ (0, 0, $dn^\infty/dz$) is applied; therefore, $n^\infty$ becomes a function of $z$, and the particle moves with a diffusiophoretic velocity $\boldsymbol{U}$. We treat the case in which the applied electrolyte concentration gradient $\nabla n^\infty$, the diffusiophoretic velocity $\boldsymbol{U}$(0, 0, $U$), and thus the liquid flow $\boldsymbol{u}(\boldsymbol{r})$ are all parallel to the $z$-axis (Figure 1). We introduce a vector $\boldsymbol{\alpha}$(0, 0, $\alpha$) proportional to $\nabla n^\infty$, viz.,

$$\boldsymbol{\alpha} = \frac{kT}{Ze}\nabla \ln(n^\infty) \tag{12}$$

　　We assume that the liquid velocity $\boldsymbol{u}(\boldsymbol{r})$ = (0, 0, $u(x, z)$) at position $\boldsymbol{r}(x, z)$ satisfies the following Navier–Stokes equations:

$$\eta\Delta\boldsymbol{u}(\boldsymbol{r}) - \nabla p(\boldsymbol{r}) - \rho_{\text{el}}(\boldsymbol{r})\nabla\psi(\boldsymbol{r}) = \boldsymbol{0} \ \text{ for } x > 0 \tag{13}$$

$$\eta\Delta\boldsymbol{u}(\boldsymbol{r}) - \gamma\boldsymbol{u}(\boldsymbol{r}) - \nabla p(\boldsymbol{r}) - \rho_{\text{el}}(\boldsymbol{r})\nabla\psi(\boldsymbol{r}) = \boldsymbol{0} \ \text{for } -d < x < 0 \tag{14}$$

with the continuity equation for $\boldsymbol{u}(\boldsymbol{r})$

$$\text{div}\boldsymbol{u}(\boldsymbol{r}) = 0 \tag{15}$$

where $p(\boldsymbol{r})$ is the pressure, and $\gamma\boldsymbol{u}(\boldsymbol{r})$ is the frictional force exerted on the liquid flow by the polymer segments in the polyelectrolyte layer, with $\gamma$ being the frictional coefficient. The boundary condition for the electric potential $\psi(\boldsymbol{r})$, far from the particle at $x\to\infty$, can be derived as follows. The ionic flows $\boldsymbol{v}_\pm(\boldsymbol{r})$, which are caused by $\boldsymbol{\alpha}$, induce a macroscopic diffusion potential field $\boldsymbol{E}$(0, 0, $E$), by which the net electric current becomes 0, and hence $\psi(\boldsymbol{r})$ tends to approach $-Ez$ as $x\to\infty$. Here, the electric current density $\boldsymbol{i}(\boldsymbol{r})$ is given by

$$\boldsymbol{i}(\boldsymbol{r}) = Ze\{n_+(\boldsymbol{r})\boldsymbol{v}_+(\boldsymbol{r}) - n_-(\boldsymbol{r})\boldsymbol{v}_-(\boldsymbol{r})\} \tag{16}$$

with

$$\boldsymbol{v}_\pm(\boldsymbol{r}) = \boldsymbol{u}(\boldsymbol{r}) - \frac{1}{\lambda_\pm}\nabla\mu_\pm(\boldsymbol{r}) \tag{17}$$

where $\boldsymbol{v}_+(\boldsymbol{r})$ and $\boldsymbol{v}_-(\boldsymbol{r})$ are the velocities of cations and anions, respectively. Since $\boldsymbol{i}(\boldsymbol{r})$ must be zero far from the particle ($x \to \infty$), we find

$$\psi(\boldsymbol{r}) \to -\beta\alpha z \ \text{ as } x \to \infty \tag{18}$$

and

$$E = \beta\alpha \tag{19}$$

where $\beta$ is defined by

$$\beta = \frac{1/\lambda_+ - 1/\lambda_-}{1/\lambda_+ + 1/\lambda_-} = -\frac{\lambda_+ - \lambda_-}{\lambda_+ + \lambda_-} \tag{20}$$

and $\alpha$ is the z-component of $\boldsymbol{\alpha}(0, 0, \alpha)$ given by (see Equation (12))

$$\alpha = \frac{kT}{Ze} \frac{d\ln(n^\infty)}{dz} \tag{21}$$

From the x-component of Equations (13) and (14), we obtain

$$p(x, z) - p(\infty, z) = 2n^\infty(z)kT\{\cosh y(x) - 1\} \tag{22}$$

where $y(x)$ is the scaled equilibrium electric potential (Equation (5)). By substituting Equation (22) into Equations (13) and (14), the z-component of Equations (13) and (14) yields

$$\frac{d^2u}{dx^2} - \frac{2Ze\alpha}{\eta}(\cosh y - 1 - \beta\sinh y) = 0 \quad \text{for } x > 0 \tag{23}$$

$$\frac{d^2u}{dx^2} - \lambda^2 u - \frac{2Ze\alpha}{\eta}(\cosh y - 1 - \beta\sinh y) = 0 \quad \text{for} -d < x < 0 \tag{24}$$

with

$$\lambda = \sqrt{\frac{\gamma}{\eta}} \tag{25}$$

where $\lambda$ is the Brinkman parameter, and $1/\lambda$ is the Brinkman screening length, which is typically of the order of 1 nm [30]. The boundary conditions for $u(x)$ are

$$u(0^-) = u(0^+) \tag{26}$$

$$\left.\frac{du}{dx}\right|_{x=0^-} = \left.\frac{du}{dx}\right|_{x=0^+} \tag{27}$$

$$u(-d) = 0 \tag{28}$$

$$u(x) \to -U \text{ as } x \to \infty \tag{29}$$

Equation (29) states that the slipping plane, at which $u(x) = 0$, is located at the particle core surface at $x = -d$. The electric potential $\psi(-d)$ is thus the zeta potential of the soft article, viz.,

$$\zeta = \psi(-d) \tag{30}$$

and

$$\tilde{\zeta} = \frac{Ze\zeta}{kT} \tag{31}$$

is the scaled zeta potential. Note that the zeta potential $\zeta = \psi(-d)$ is different from $\psi_0 = \psi(0)$, i.e., the electric potential at the front edge of the polyelectrolyte layer at $x = 0$, which we call the surface potential of the soft particle.

We introduce the scaled diffusiophoretic mobility $U^*$ by

$$\boldsymbol{U} = \frac{\varepsilon_r \varepsilon_0}{\eta}\left(\frac{kT}{Ze}\right)U^* \boldsymbol{\alpha} \tag{32}$$

By solving Equations (23) and (24) subject to Equations (26)–(29), we obtain

$$U^* = 4\ln\left[\cosh\left(\frac{y_0}{4}\right)\right] + \beta y_0 + \frac{2\kappa}{\lambda}\tanh(\lambda d)\left\{\cosh\left(\frac{y_0}{2}\right) - 1 + \beta\sinh\left(\frac{y_0}{2}\right)\right\} + \frac{\kappa^2}{\lambda}\int_{-d}^{0}(\cosh y - 1 + \beta\sinh y)\frac{\sinh[\lambda(x+d)]}{\cosh(\lambda d)}dx \tag{33}$$

with

$$y_o = \frac{Ze\psi_o}{kT} \tag{34}$$

where $y_o = y(0)$ is the scaled surface potential of the soft particle.

## 3. Results and Discussion

Equation (33) is the general expression of the diffusiophoretic mobility $U^*$ of a large soft particle. Consider some limiting cases for Equation (33). For $d = 0$, in which case the polyelectrolyte layer vanishes, the soft particle becomes a hard particle with no polyelectrolyte layer carrying zeta potential $\zeta = \psi_o = \psi(0) = \psi(-d)$, and Equation (33) becomes

$$U^* = 4\ln\left[\cosh\left(\frac{\widetilde{\zeta}}{4}\right)\right] + \beta\widetilde{\zeta} \tag{35}$$

which agrees with the diffusiophoretic mobility of a large hard particle with zeta potential $\zeta$ [1,2]. For $\lambda \to 0$ and $\rho_{\text{fix}} \to 0$, in which case the polyelectrolyte layer also vanishes, the soft particle becomes a hard particle carrying zeta potential $\zeta = \psi(-d)$, and Equation (33) becomes Equation (35). For the limit of $\lambda \to \infty$, the soft particle becomes a hard particle carrying surface potential $\zeta = \psi_o$, and Equation (34) tends to

$$U^* = 4\ln\left[\cosh\left(\frac{y_o}{4}\right)\right] + \beta y_o \tag{36}$$

which is the diffusiophoretic mobility of a large rigid particle with a surface located at $x = -d$. In this case, there is no liquid flow inside the polyelectrolyte layer, i.e., the slipping plane shifts to a plane $x = 0$, but electrolyte ions can penetrate the polyelectrolyte layer.

Now, in the following, we consider three important cases in detail.

### 3.1. Weakly Charged Soft Particle

We first consider the low potential case. In this case, Equation (33) becomes

$$U^* = \beta y_o + \frac{y_o^2}{8} + \frac{\kappa}{\lambda}\tanh(\lambda d)\left(\beta y_o + \frac{y_o^2}{4}\right) + \frac{\kappa^2}{\lambda}\int_{-d}^{0}\left\{\beta y(x) + \frac{y^2(x)}{2}\right\}\frac{\sinh[\lambda(x+d)]}{\cosh(\lambda d)}dx \tag{37}$$

with

$$y(x) = \widetilde{\zeta}e^{-\kappa(x+d)} + y_{\text{DON}}\left\{1 - e^{-\kappa d}\cosh[\kappa(x+d)]\right\} \tag{38}$$

$$\zeta = \psi(-d) = \frac{\sigma}{\varepsilon_r\varepsilon_o\kappa} \tag{39}$$

$$\widetilde{\zeta} = y(-d) = \frac{Ze\zeta}{kT} \tag{40}$$

$$\psi_{\text{DON}} = \frac{\rho_{\text{fix}}}{\varepsilon_r\varepsilon_o\kappa^2} \tag{41}$$

$$y_{\text{DON}} = \frac{\rho_{\text{fix}}}{\varepsilon_r\varepsilon_o\kappa^2}\left(\frac{Ze}{kT}\right) \tag{42}$$

where Equation (39) is the relation between $\zeta$ and $\sigma$ for the low potential case, $\psi_{\text{DON}}$ is the Donnan potential (Equation (7)) for the low potential case, and $y_{\text{DON}}$ is its scaled quantity. By substituting Equation (38) into Equation (37), we obtain

$$U^* = \beta y_o + \frac{y_o^2}{8} + \frac{\kappa}{\lambda}\tanh(\lambda d)\left(\beta y_o + \frac{y_o^2}{4}\right) + \beta\left(f_1\widetilde{\zeta} + f_2 y_{\text{DON}}\right) + f_3\widetilde{\zeta}^2 + f_4 y_{\text{DON}}^2 + f_5\widetilde{\zeta}y_{\text{DON}} \tag{43}$$

where $f_1$–$f_5$ are defined by

$$f_1 = \frac{\kappa^2}{\kappa^2 - \lambda^2}\left[\frac{1}{\cosh(\lambda d)} - \left\{1 + \frac{\kappa}{\lambda}\tanh(\lambda d)\right\}e^{-\kappa d}\right] \tag{44}$$

$$f_2 = \frac{\kappa^2(2\kappa^2 - \lambda^2)}{2\lambda^2(\kappa^2 - \lambda^2)} - \frac{\kappa^2}{\lambda^2\cosh(\lambda d)} - \frac{\kappa^3\tanh(\lambda d)}{2\lambda(\kappa^2 - \lambda^2)} - \frac{\kappa^2 e^{-\kappa d}}{(\kappa^2 - \lambda^2)\cosh(\lambda d)} + \frac{\kappa^2\{\lambda + \kappa\tanh(\lambda d)\}e^{-2\kappa d}}{2\lambda(\kappa^2 - \lambda^2)} \tag{45}$$

$$f_3 = \frac{\kappa^2}{2(4\kappa^2 - \lambda^2)}\left[\frac{1}{\cosh(\lambda d)} - \left\{1 + \frac{2\kappa}{\lambda}\tanh(\lambda d)\right\}e^{-2\kappa d}\right] \tag{46}$$

$$f_4 = \frac{\kappa^2}{8\lambda^2} - \frac{\kappa^2}{2\lambda^2\cosh(\lambda d)} - \frac{\kappa^3\tanh(\lambda d)}{4\lambda(4\kappa^2-\lambda^2)} + \frac{3\kappa^5\{\kappa-\lambda\tanh(\lambda d)\}}{2\lambda^2(\kappa^2-\lambda^2)(4\kappa^2-\lambda^2)} - \frac{\kappa^2 e^{-\kappa d}}{(\kappa^2-\lambda^2)\cosh(\lambda d)}$$
$$+ \frac{\kappa^2\{\kappa^2+\lambda^2+2\kappa\lambda\tanh(\lambda d)\}e^{-2\kappa d}}{4\lambda^2(\kappa^2-\lambda^2)} - \frac{\kappa^2(2\kappa^2-\lambda^2)e^{-2\kappa d}}{2\lambda^2(4\kappa^2-\lambda^2)\cosh(\lambda d)} - \frac{\kappa^2\{\lambda+2\kappa\tanh(\lambda d)\}e^{-4\kappa d}}{8\lambda(4\kappa^2-\lambda^2)} \tag{47}$$

$$f_5 = \frac{\kappa^2}{(\kappa^2 - \lambda^2)\cosh(\lambda d)} + \frac{\kappa^2(2\kappa^2 - \lambda^2)e^{-\kappa d}}{\lambda^2(4\kappa^2 - \lambda^2)\cosh(\lambda d)} - \frac{\kappa^2\{\kappa^2 + \lambda^2 + 2\kappa\lambda\tanh(\lambda d)\}e^{-\kappa d}}{2\lambda^2(\kappa^2 - \lambda^2)} + \frac{\kappa^2\{\lambda + 2\kappa\tanh(\lambda d)\}e^{-3\kappa d}}{2\lambda(4\kappa^2 - \lambda^2)} \tag{48}$$

### 3.2. Soft Particle with a Thick Polyelectrolyte Layer

We next consider a soft particle with $\kappa d \gg 1$ and $\lambda d \gg 1$, which are fulfilled for most practical cases in biological systems. In such a case, the contribution of the surface charge of the particle core with density $\sigma$ can be neglected, and the electric potential far inside the polyelectrolyte layer is practically equal to the Donnan potential $\psi_{\text{DON}}$ (Equation (7)). We may thus linearize Equation (3) with respect to the deviation of the potential $y(x)$ from $y_{\text{DON}}$ with the result [27–32]

$$y(x) = y_{\text{DON}} + (y_{\text{o}} - y_{\text{DON}})e^{\kappa_{\text{m}}x} \text{ for } -d < x < 0 \tag{49}$$

with

$$\kappa_{\text{m}} = \kappa\sqrt{\cosh y_{\text{DON}}} \tag{50}$$

$$y_{\text{o}} = y(0) = y_{\text{DON}} - \tanh\left(\frac{y_{\text{DON}}}{2}\right) = \ln\left[\frac{\rho_{\text{fix}}}{2Zen^{\infty}} + \sqrt{\left(\frac{\rho_{\text{fix}}}{2Zen^{\infty}}\right)^2 + 1}\right] + \frac{2Zen^{\infty}}{\rho_{\text{fix}}}\left\{1 - \sqrt{\left(\frac{\rho_{\text{fix}}}{2Zen^{\infty}}\right)^2 + 1}\right\} \tag{51}$$

where $\kappa_{\text{m}}$ is the effective Debye–Hückel parameter in the polyelectrolyte layer. By using Equation (49) and noting that the lower limit $-d$ of integration in Equation (33) can be replaced with $-\infty$ for $\kappa d \gg 1$ and $\lambda d \gg 1$, we finally obtain the following approximate expression for $U^*$:

$$U^* = 4\ln\left[\cosh\left(\frac{y_{\text{o}}}{4}\right)\right] + \frac{4\kappa}{\lambda}\sinh^2\left(\frac{y_{\text{o}}}{4}\right) + \frac{\kappa_{\text{m}}(\kappa_{\text{m}}^2 - \kappa^2)}{\lambda^2(\kappa_{\text{m}} + \lambda)} + \beta\left\{\frac{y_{\text{o}}/\kappa_{\text{m}} + y_{\text{DON}}/\lambda}{1/\kappa_{\text{m}} + 1/\lambda} + \frac{\kappa^2}{\lambda^2}\sinh(y_{\text{DON}})\right\} \tag{52}$$

The term proportional to $\beta$ resulting from the diffusion potential field corresponds to the electrophoretic mobility $\mu$ of a large soft particle with $\kappa d \gg 1$ and $\lambda d \gg 1$, which is given by [28–31]:

$$\mu = \frac{\varepsilon_{\text{r}}\varepsilon_{\text{o}}}{\eta} \cdot \frac{\psi_{\text{o}}/\kappa_{\text{m}} + \psi_{\text{DON}}/\lambda}{1/\kappa_{\text{m}} + 1/\lambda} + \frac{\rho_{\text{fix}}}{\eta\lambda^2} \tag{53}$$

As $\kappa \to \infty$, $U^*$ tends to a non-zero constant value independent of the electrolyte concentration, viz.,

$$U^* = \beta\frac{\kappa^2}{\lambda^2}\sinh(y_{\text{DON}}) = \beta\frac{\rho_{\text{fix}}}{\lambda^2\varepsilon_{\text{r}}\varepsilon_{\text{o}}}\left(\frac{Ze}{kT}\right) \tag{54}$$

This is an electrokinetic characteristic of soft particles [28–31].

For the low potential case, Equation (52) tends to

$$U^* = \beta\left(\frac{\kappa + \lambda/2}{\kappa + \lambda} + \frac{\kappa^2}{\lambda^2}\right)y_{\text{DON}} + \frac{1}{32}\left\{1 + \frac{2\kappa}{\lambda} + \frac{16\kappa^3}{\lambda^2(\kappa + \lambda)}\right\}y_{\text{DON}}^2 \tag{55}$$

which is different from the correct limiting form of Equation (44) for $\kappa d \gg 1$ and $\lambda d \gg 1$, viz.,

$$U^* = \beta\left(\frac{\kappa + \lambda/2}{\kappa + \lambda} + \frac{\kappa^2}{\lambda^2}\right)y_{DON} + \frac{1}{32}\left\{1 + \frac{2\kappa}{\lambda} + \frac{16\kappa^3}{\lambda^2(\kappa + \lambda)} + \frac{2\kappa^2}{\lambda(\kappa + \lambda/2)}\right\}y_{DON}^2 \quad (56)$$

which agrees with Equation (24) in a previous paper [26]. By comparing Equations (55) and (56), we see that Equation (54) fails to reproduce the last term in the curly brackets in Equation (56). This is because this term is obtained by the integrating of terms of the order of $(y_o - y_{DON})^2$, which is neglected in Equation (49). Numerically, however, this error is small, since the dominant term in the curly brackets in Equation (56) is the third term when $\kappa/\lambda$ is small, and it is the first term when $\kappa/\lambda$ is large.

Figure 2 shows some examples of the results of the calculation of the scaled diffusiophoretic velocity $U^*$ of a soft particle with a thick polyelectrolyte layer in an aqueous KCl solution ($\beta = -0.02$) and in an aqueous NaCl solution ($\beta = -0.2$) as a function of the scaled Donnan potential $y_{DON}$ for various values of the ratio $\kappa/\lambda$ of the Brinkman shielding length $1/\lambda$ to the Debye length $1/\kappa$, calculated with Equation (52) (solid lines). Results obtained with a low potential approximation with Equation (56) are also given (dotted lines) in Figure 2, which shows to be a good approximation for $|y_{DON}| \leq 1$.

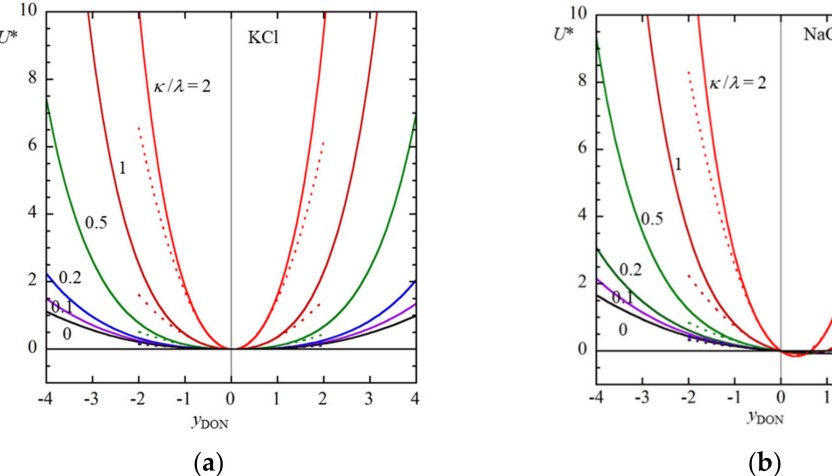

(a)　　　　　　　　　　　　　　(b)

**Figure 2.** Scaled diffusiophoretic mobility $U^*$ of a large soft particle with a thick polyelectrolyte layer as a function of the scaled Donnan potential $y_{DON}$ for several values of $\kappa/\lambda$ in an aqueous KCl solution ($\beta = -0.02$) (**a**) and in an aqueous NaCl solution ($\beta = -0.2$) (**b**). Solid lines are results obtained with Equation (52). Results obtained with a low potential approximation with Equation (56) are also given as dotted lines.

### 3.3. Soft Particle with an Uncharged Polymer Layer

Finally, we consider a soft particle with $\rho_{fix} = 0$. This case corresponds to a soft particle covered with an uncharged polymer layer. The solution to the Poisson–Boltzmann equations (Equations (2) and (3)), in this case, is found to be

$$y(x) = 2\ln\left[\frac{1 + \tanh\left(\tilde{\zeta}/4\right)e^{-\kappa(x+d)}}{1 - \tanh\left(\tilde{\zeta}/4\right)e^{-\kappa(x+d)}}\right] \quad \text{for } x \geq -d \quad (57)$$

which is equivalent to

$$\tanh\left[\frac{y(x)}{4}\right] = \tanh\left(\frac{\tilde{\zeta}}{4}\right)e^{-\kappa(x+d)} \text{ for } x \geq -d \quad (58)$$

By using Equation (60), we obtain

$$\sinh[y(x)] = \sum_{n=1}^{\infty} \frac{(2n-1)}{(2n-1)^2 - (\lambda/\kappa)^2} \tanh^{2n-1}\left(\frac{\widetilde{\zeta}}{4}\right) \tag{59}$$

$$\cosh[y(x)] - 1 = \sum_{n=1}^{\infty} \frac{(2n)}{(2n)^2 - (\lambda/\kappa)^2} \tanh^{2n}\left(\frac{\widetilde{\zeta}}{4}\right) \tag{60}$$

By substituting Equations (59) and (60) into Equation (33) with $\rho_{\text{fix}} = 0$, we obtain

$$\begin{aligned}
U^* &= 4\ln\left[\cosh\left(\frac{y_o}{4}\right)\right] + \beta y_o + \frac{2\kappa}{\lambda}\tanh(\lambda d)\left\{\cosh\left(\frac{y_o}{2}\right) - 1 + \beta\sinh\left(\frac{y_o}{2}\right)\right\} \\
&+ 4\sum_{n=1}^{\infty} \frac{(2n)}{(2n)^2 - \left(\frac{\lambda}{\kappa}\right)^2}\tanh^{2n}\left(\frac{\widetilde{\zeta}}{4}\right)\left[\frac{1}{\cosh(\lambda d)} - \left\{1 + (2n)\frac{\kappa}{\lambda}\tanh(\lambda d)\right\}e^{-2n\kappa d}\right] \\
&+ 4\beta\sum_{n=1}^{\infty} \frac{(2n-1)}{(2n-1)^2 - (\lambda/\kappa)^2}\tanh^{2n-1}\left(\frac{\widetilde{\zeta}}{4}\right)\left[\frac{1}{\cosh(\lambda d)} - \left\{1 + (2n-1)\frac{\kappa}{\lambda}\tanh(\lambda d)\right\}e^{-(2n-1)\kappa d}\right]
\end{aligned} \tag{61}$$

with

$$y_o = y(0) = 2\ln\left[\frac{1 + \tanh\left(\widetilde{\zeta}/4\right)e^{-\kappa d}}{1 - \tanh\left(\widetilde{\zeta}/4\right)e^{-\kappa d}}\right] \tag{62}$$

which becomes Equation (39) for the case of a weakly charged soft particle. The scaled zeta potential $\widetilde{\zeta}$ is related to $\sigma$ by

$$\frac{\sigma}{\varepsilon_r\varepsilon_o\kappa}\left(\frac{Ze}{kT}\right) = 2\sinh\left(\frac{\widetilde{\zeta}}{2}\right) \tag{63}$$

As $\kappa d \to 0$, Equation (61) tends to

$$U^* = 4\ln\left[\cosh\left(\frac{\widetilde{\zeta}}{4}\right)\right] + \beta\widetilde{\zeta} \tag{64}$$

whereas, as $\kappa d \to \infty$, Equation (61) tends to

$$U^* = \left\{4\ln\left[\cosh\left(\frac{\widetilde{\zeta}}{4}\right)\right] + \beta\widetilde{\zeta}\right\}\frac{1}{\cosh(\lambda d)} \tag{65}$$

For the case of a weakly charged soft particle, Equation (61) becomes

$$U^* = \beta y_o + \frac{y_o^2}{8} + \frac{\kappa}{\lambda}\tanh(\lambda d)\left(\beta y_o + \frac{y_o^2}{4}\right) + \beta f_1\widetilde{\zeta} + f_3\widetilde{\zeta}^2 \tag{66}$$

with

$$y_o = y(0) = \widetilde{\zeta}e^{-\kappa d} \tag{67}$$

Equation (67) agrees with Equation (43) with $\rho_{\text{fix}} = 0$.

Figures 3 and 4 show some examples of the results of the calculation of the scaled diffusiophoretic velocity $U^*$ of a soft particle in an aqueous KCl solution ($\beta = -0.02$) (Figure 3) and in an aqueous NaCl solution ($\beta = -0.2$) (Figure 4) as a function of the scaled zeta potential $\widetilde{\zeta}$ for several values of $\kappa d$ at $\lambda d = 1$ (a) and $\lambda d = 10$ (b), calculated with Equation (61) (solid lines). Results obtained with a low potential approximation with Equation (66) are also given (dotted lines) in Figure 3, which shows to be a good approximation for $|\widetilde{\zeta}| \leq 2$.

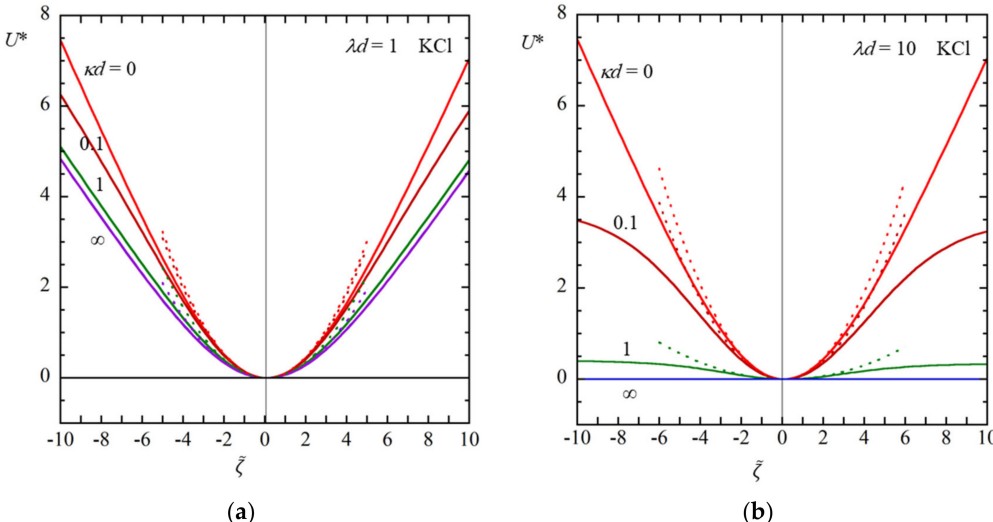

**Figure 3.** Scaled diffusiophoretic mobility $U^*$ of a large soft particle as a function of the scaled zeta potential $\widetilde{\zeta}$ for several values of $\kappa d$ at $\lambda d = 1$ (**a**) and $\lambda d = 10$ (**b**) in an aqueous KCl solution ($\beta = -0.02$). Solid lines are results obtained with Equation (61). Results obtained with a low potential approximation with Equation (66) are also given as dotted lines.

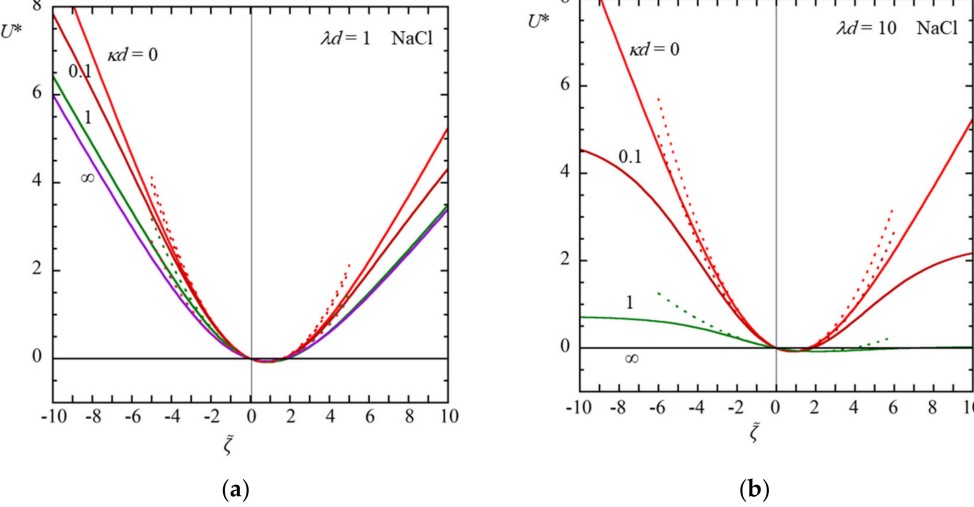

**Figure 4.** Scaled diffusiophoretic mobility $U^*$ of a large soft particle as a function of the scaled zeta potential $\widetilde{\zeta}$ for several values of $\kappa d$ at $\lambda d = 1$ (**a**) and $\lambda d = 10$ (**b**) in an aqueous NaCl solution ($\beta = -0.2$). Solid lines are results obtained with Equation (61). Results obtained with a low potential approximation with Equation (66) are also given as dotted lines.

## 4. Conclusions

We have derived the general expression (Equation (33)) for the diffusiophoretic mobility $U^*$ of a soft particle carrying $\rho_{\text{fix}}$ and $\sigma$ of arbitrary values in an electrolyte concentration gradient for the case in which the particle size is large enough compared with the Debye length, $1/\kappa$. Therefore, the particle surface can be regarded as planar. We have derived approximate mobility expressions for the following three cases: (i) a weakly charged soft particle (Equation (43)), (ii) a soft particle with a thick polyelectrolyte layer (Equation (52)), in which the thickness of the polyelectrolyte layer is much larger than the Debye length $1/\kappa$ and the Brinkman screening length $1/\lambda$, and (iii) a soft particle with an uncharged polymer layer (Equation (61)). It has been shown that, for case (ii) for the limit of a high electrolyte concentration, diffusiophoretic mobility tends to a non-zero limiting value given by Equation (54), independent of the electrolyte concentration.

**Funding:** This research received no external funding.

**Institutional Review Board Statement:** Not applicable.

**Informed Consent Statement:** Not applicable.

**Data Availability Statement:** Not applicable.

**Conflicts of Interest:** The author declares no conflict of interest.

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
