# Peer review of "Diffusiophoresis of a Soft Particle as a Model for Biological Cells"

_colloids, doi:10.3390/colloids6020024_

Round 1

Reviewer 1 Report

In this work, the author develops an expression for the diffusiophoretic velocity (that is, the velocity promoted by electrolyte concentration gradients) of large soft particles with an arbitrary charge. They also provide analytical expressions for three limiting cases: low particle charge, thick soft layer and uncharged layer. The manuscript is well written and the description is clear. The behavior of soft particles in different media is a current relevant topic in many fields. The effort of developing analytical expression has the added value of being accessible to other researcher, and hence I’m very positive on this manuscript and I recommend it for publication. In order to increase the relevance of this work, I suggest to add some comments about the relevance of diffusiophoresis in microfluidic devices, where it exists a big interest in strategies for controlling the fluid and particles dynamics inside.

Author Response

Thank you very much for your comment. We have added the sentence “In addition, as Shin [34] has pointed out, there is a growing interest in microfluidic colloid separation enabled by diffusiophoresis in various biological and biomedical applications.” and added Ref. [34]. 

Reviewer 2 Report

Professor Ohshima is a highly respected scholar in the field of electrokinetics. He is the author of the classic book “Theory of Colloid and Interfacial Electric Phenomena,” Elsevier, 2006. This is yet another excellent work by Professor Ohshima, and I recommend publication of the manuscript as is.

Author Response

Thank you very much for your recommendation "Accept as is".

Reviewer 3 Report

I have attached my report as a pdf. 

Author Response

Comment 1: The title says that the model is for biological cells. I am aware that the model does not directly reflect actual features of the cells, but since there are experimental studies that demonstrated diffusiophoresis of cells, maybe mentioning potential relevance to those studies (e.g. Hartman et al. New Biotechnology 2018 and Shim et al. Soft Matter 2021) in the introduction can be helpful for some readers.

Reply: Thank you very much for your comment. We have added the sentence “There are several experimental studies on the diffusiophoresis of biological cells [32, 33]” in the Introduction section and added References

Hartman et al. New Biotechnology 2018 and Shim et al. Soft Matter 2021 as Refs. [32, 33]

Comment 2: On the build-up of the theory, it sounds a little awkward to read “concentration gradient ∇ ln(? ∞) is set up”. If there is a concentration field ? ∞ in the bulk, its gradient is ∇? ∞, and the charge neutrality with the zero current condition yields the electric field ? = − ?? ?? = ? ?? ?? ? ln(?∞) ?? near the surface. The mathematical description in the article leads to the same statements (eq 16-19), but better physical description can be provided since the functional form ln(? ∞) of the bulk concentration is not required to obtain such electric field.

Reply: Thank you very much for your comment. We have changed ∇ ln(? ∞) into ∇? ∞ in the relevant sentences in the revised manuscript.

Comment 3: On the definition of the Brinkman parameter (eq 25), could you mention typical scales for a known polyelectrolyte (just like what you did for typical cell size and Debye length in the introduction)?

Reply: Thank you very much for your comment. We have added the following sentence after Eq. (25): “which is typically of the order of 1 nm [30]”,

Comment 4: KCl represents one of the common biological salts and it has small ? (so chemiphoresis is dominant). It will be helpful to add one more salt that has higher electrophoretic contribution (e.g. NaCl, LiCl,..) in the model study to investigate diffusiophoretic behaviors of different particles (or regimes) near small yDON.

Reply: Thank you very much for your valuable comment. We have added Figure 2(b), Figure 4(a), and Figure 4(b) for the diffusiophoretic mobility for the case of NaCl in the revised manuscript.